# Deciphering Plant-Insect-Microorganism Signals for Sustainable Crop Production

**DOI:** 10.3390/biom13060997

**Published:** 2023-06-15

**Authors:** Gareth Thomas, Quint Rusman, William R. Morrison, Diego M. Magalhães, Jordan A. Dowell, Esther Ngumbi, Jonathan Osei-Owusu, Jessica Kansman, Alexander Gaffke, Kamala Jayanthi Pagadala Damodaram, Seong Jong Kim, Nurhayat Tabanca

**Affiliations:** 1Protecting Crops and the Environment, Rothamsted Research, Harpenden, AL5 2JQ, UK; 2Department of Systematic and Evolutionary Botany, University of Zürich, Zollikerstrasse 107, 8008 Zürich, Switzerland; quint.rusman@systbot.uzh.ch; 3United States Department of Agriculture-Agricultural Research Service (USDA-ARS), Center for Grain and Animal Health Research, 1515 College Ave., Manhattan, KS 66502, USA; william.morrison@usda.gov; 4Luiz de Queiroz College of Agriculture, University of São Paulo, Piracicaba 13418-900, SP, Brazil; magalhaes.dmm@gmail.com; 5Department of Plant Sciences, University of California, Davis, One Shields Ave., Davis, CA 95616, USA; jordan.dowell@gmail.com; 6Department of Entomology, University of Illinois at Urbana Champaign, Urbana, IL 61801, USA; enn@illinois.edu; 7Department of Biological, Physical and Mathematical Sciences, University of Environment and Sustainable Development, Somanya EY0329-2478, Ghana; josei-owusu@uesd.edu.gh; 8Center for Chemical Ecology, Department of Entomology, The Pennsylvania State University, University Park, PA 16802, USA; kansmanj@psu.edu; 9United States Department of Agriculture-Agricultural Research Service (USDA-ARS), Center for Medical, Agricultural, and Veterinary Entomology, 6383 Mahan Dr., Tallahassee, FL 32308, USA; alexander.gaffke@usda.gov; 10Division of Crop Protection, ICAR-Indian Institute of Horticultural Research, Hesseraghatta Lake PO, Bangalore 560089, India; kamalajayanthi.pd@icar.gov.in; 11United States Department of Agriculture-Agricultural Research Service (USDA-ARS), Natural Products Utilization Research Unit, University, MS 38677, USA; seong.kim@usda.gov; 12United States Department of Agriculture-Agricultural Research Service (USDA-ARS), Subtropical Horticulture Research Station, 13601 Old Cutler Rd., Miami, FL 33158, USA

**Keywords:** chemical ecology, semiochemicals, volatile organic compounds, kairomones, pheromones, biocontrol, plants, insects, microbes

## Abstract

Agricultural crop productivity relies on the application of chemical pesticides to reduce pest and pathogen damage. However, chemical pesticides also pose a range of ecological, environmental and economic penalties. This includes the development of pesticide resistance by insect pests and pathogens, rendering pesticides less effective. Alternative sustainable crop protection tools should therefore be considered. Semiochemicals are signalling molecules produced by organisms, including plants, microbes, and animals, which cause behavioural or developmental changes in receiving organisms. Manipulating semiochemicals could provide a more sustainable approach to the management of insect pests and pathogens across crops. Here, we review the role of semiochemicals in the interaction between plants, insects and microbes, including examples of how they have been applied to agricultural systems. We highlight future research priorities to be considered for semiochemicals to be credible alternatives to the application of chemical pesticides.

## 1. Introduction

Pests and pathogens pose a major constraint to agricultural food production by reducing the yield and quality of crops. Yield losses due to pests and pathogens range from 20–80% in crops such as wheat, rice, cowpea, and soybean [1,2]. Climate warming could expand the geographic distribution of insect pests, increasing both the likelihood of invasive pest species introduction and damage and the incidence of insect-transmitted plant diseases [3]. Similarly, yield losses caused by pathogens are also expected to grow with increases in pathogen abundance associated with global temperature rise and climate change (reviewed in [4]). This particularly seems to be the case for soil fungal pathogens, which demonstrate a worldwide increase in relative abundance with rising global temperatures [5]. The combination of global temperature rises and anticipated population growth increases the complexity of achieving global food security, for which more sustainable pest and pathogen management strategies are required.

The application of chemical pesticides has been integral to reducing yield loss from pests and pathogens in post-green revolution agriculture. However, future development and use of chemical pesticides is unsustainable. The development of resistance by target pests can occur due to pesticide over-application, which renders them less effective, with losses due to resistance estimated at $10 billion per year in the USA alone [6]. Moreover, the costs of bringing a new pesticide to market are increasing [7,8]. Sustainable alternatives to these chemical pesticides are therefore required, including semiochemicals and small lipophilic molecules used by organisms to communicate intraspecifically (pheromones) or interspecifically (allelochemicals) within and across trophic levels [9]. Detection of semiochemicals from a producing organism alters the behavioural and/or developmental processes of the receiving organism [10,11]. Compared to pesticides (e.g., glycophosphates, dichlorodiphenyltrichloroethane, azole-fungicides, etc.), semiochemicals are generally less toxic and readily break down in the environment, thereby offering a promising crop protection tool for sustainable agriculture. In nature, semiochemicals are integral to plant, insect, and microbial interactions, where the composition and quantity of compounds emitted can elicit different responses (Figure 1) [12].

This review will provide an overview of how semiochemical signalling between plants, insects and microbes could be used to improve agricultural sustainability, based in part on the “Early Career Symposium: Deciphering Plant-Insect-Microorganism Signals for Sustainable Crop Production”, as well as the authors’ own perspectives on the field of chemical ecology. The symposium was held at the 2022 Fall National Meeting of the American Chemical Society (ACS), in the Agrochemical Division of ACS, with co-sponsorship from the Division of Agricultural and Food Chemistry, Committee on Technician Affairs and Division of Biochemical Technology. The role of semiochemicals produced by plants, insects, and microbes will be discussed, focusing on leveraging semiochemicals to improve agricultural sustainability and suggesting future research priorities. The examples we have selected are based on more recent studies, whilst also including earlier examples where certain phenomena were first established to provide broader context.

## 2. Plant-Insect Signals

Insects rely on semiochemicals when searching for suitable hosts, mates, and oviposition sites, establishing a territory and escaping competition [13,14,15]. In the host plant discrimination process, insects use olfactory receptor neurons in the sensilla of their antennae to recognize and distinguish individual molecules emitted by plants in a complex background of VOCs [12]. Whilst single compounds can mediate plant-insect interactions, most interactions between plants and insects are mediated by blends containing several compounds [12,16,17]. Moreover, plant VOC emissions can vary over space, time, and environmental context, making reliance on single compounds a poor strategy for insects [18,19,20,21,22]. These complex blends provide more ways to discern potential hosts and their quality. Delineating insect use of complex host-semiochemical blends to balance repelling pests and attracting beneficial insects is a challenging opportunity for agricultural innovation. This section will focus on examples from more recent literature across a broad range of ecological processes encompassing plant-insect chemical interactions.

### 2.1. Plant VOCs Induction by Insect Pests

Based on our understanding of the dynamic nature of insect foraging behaviour and the role of constitutive and induced plant VOCs, plant selection by crop-associated insects can be altered directly and indirectly. Crop-associated insects include antagonistic pests that feed on plants (herbivores), mutualistic visitors that aid plant reproduction (e.g., pollinators), predators that feed on herbivores, and parasitoids.

Herbivores, predators, and pollinators use VOCs emitted constitutively from vegetative and flowering plants to locate their preferred crop or pest insect feeding on those crops. Moreover, oviposition and herbivory can induce changes in plant VOC emissions, termed oviposition-induced plant volatiles (OIPVs) [23] and herbivore-induced plant volatiles (HIPVs) [24], respectively. OIPVs and HIPVs are used by predators as a strategy to locate herbivore-infested plants [24,25,26,27]. In addition, herbivores and pollinators can use such signals to avoid or locate damaged plants [24,28,29]. Certain predators feed on plant tissue as well as their prey (zoophytophagous predators), which can also induce plant volatiles, termed zoophytophagous-induced plant volatiles (ZIPVs) [30]. ZIPVs can repel herbivores and attract conspecific predators, as well as parasitoids of herbivores [31]. The direct alteration of host plant selection by insects via semiochemicals can be done by manipulating plant production of key compounds or the ratios of key compounds, which can be performed by external application of natural or synthetic compounds or obtaining cultivars with altered VOC emissions via breeding or genetic engineering.

### 2.2. Exogenous Application of Natural or Synthetic Compounds

Exogenous application of natural or synthetic compounds can either alter insect host plant selection directly or alter crop plant VOC emission, thereby indirectly altering insect host plant selection. For example, slow-release beads containing a synthetic mixture of two compounds ((*E*)-β-farnesene and methyl salicylate) reduced aphid abundance and increased parasitism rates in wheat fields [32]. (*E*)-β-Farnesene is released by aphids as an alarm pheromone [33] but can also be found in the scent of flowers [34]. Methyl salicylate, which is attractive to a variety of predatory insects [35], is the methylated form of the plant hormone salicylic acid that is synthesized in response to a range of phytopathogens [36]. VOC dispensers containing a synthetic mixture of four HIPV compounds ((*Z*)-3-hexenyl acetate, α-pinene, sabinene, and *n*-heptanal) increased the incidence of the parasitic wasp *Cotesia vestalis* in *Brassica* crops, resulting in the reduced incidence of the diamondback moth *Plutella xylostella*, a pest of cruciferous crops, under both greenhouse [37] and field conditions [38]. A mixture of geraniol, citral, anethole, and linalool increased the yield of red clover, likely due to enhanced pollinator attraction [39]. For more examples of synthetic HIPVs, see Ayelo et al., 2021 [30]. Whilst an exogenous application can offer benefits to crops through altering insect host plant selection, caution should be expressed with the application of floral scent compounds, as such compounds can attract flower-feeding pest insects as well as pollinators [40,41]. For example, in an attempt to increase pollinator attraction, Theis and Adler (2012) enhanced the floral scent of wild Texas gourd, *Cucurbita pepo* var. *texana*, with the dominant compound of its scent bouquet, 1,4-dimethoxybenzene [42]. Rather than attracting more pollinators, plants with scent emitters attracted more florivorous striped cucumber beetles, *Acalymma vittatum*, resulting in reduced seed production. The exogenous application of natural or synthetic compounds can also be applied to honeybee hives. Feeding honeybee colonies with a sucrose solution scented with an apple or pear mimic odour enhanced bee foraging and pollination activities in apple and pear crops [43]. Exogenous application of plant defence activators also seems promising for altering crop VOC emission and insect host plant selection [24]. The application of synthetic *(Z)*-jasmone, a known plant defence activator, has been shown to alter VOC production across a range of crops [44,45,46,47,48,49]. Hence, direct alterations of host plant selection behaviour of crop-associated insects can be used to repel antagonistic herbivorous pest insects and attract mutualistic pollinators and predators.

### 2.3. Crop Genetic Diversity and VOC Emissions

In addition to the external manipulation of crop-associated insect behaviour, crop genetic diversity underpins constitutive and induced VOC emissions. Cultivars of modern crop varieties and their wild relatives vary in their constitutive and inducible VOC emissions due to modifications in plant defence characteristics following crop domestication [50]. This is important for developing new varieties with altered host cues and induced VOC defences. Indeed, many compounds that plants emit constitutively or when under attack by herbivores are toxic or repellent towards the herbivores [51].

Altering the VOCs emitted by crops that are used in host plant locations by herbivores can be achieved in multiple ways. Breeding targets can aim to decrease the emission of attractive VOCs used by herbivores in host plant recognition or quality assessment (positive stimuli) or increase the emission of VOCs that repel herbivores (negative stimuli). Importantly, the ratio of specific compounds can have an influence on the attractant/repellent properties of the plant. For example, transgenic grapevine, *Vitis vinifera*, with altered emission in the ratio of two compounds, (*E*)-β-caryophyllene and (*E*)-β-farnesene, was shown to be less attractive to the European grapevine moth *Lobesia botrana*, compared to extracts sampled from control (wild-type) plants [52]. This demonstrates that altering the ratios of plant VOC production can disrupt the host location of insect herbivores.

To increase host location by natural enemies of insect herbivores, cultivars with greater HIPV responses are desirable, although extensive variation in HIPV blends can occur across cultivars and wild relatives ([53] and references therein). In extreme examples, the HIPV blend of certain modern cultivars has lost its defensive functions. The HIPV blend of these modern maize hybrid cultivars neither repelled new colonising corn leafhoppers, *Dalbulus maidis*, nor attracted their natural enemy [54]. This genetic and phenotypic variation poses challenges for consistent biological control but also supplies extensive genetic variation, which breeders can implement into breeding programmes. Certain compounds, such as methyl salicylate, are attractive to a wide range of natural enemies and can be implemented into HIPV blends to make them more attractive. Importantly, recent advances in our understanding of HIPVs and plant-herbivore interactions suggest that breeding targets for HIPVs must consider more than the quantity and quality of the HIPV blend. Certain pest insects, such as *Spodoptera frugiperda* caterpillars, can suppress HIPV emission in maize, although the plant’s attractiveness to the parasitoid wasp *Cotesia marginiventris* was not affected by HIPV suppression [55]. Moreover, HIPVs induced by one pest can provide enemy-free space for a secondary pest, reducing the chances that their offspring are attacked by natural enemies. An example of this involves two destructive pests of rice; the striped stem borer, *Chilo suppressalis*, as the primary pest and the brown planthopper, *Nilaparvata lugens*, as a secondary pest; the egg parasitoid *Anagrus nilaparvatae* locates *N. lugens*-infested plants through HIPVs [56,57]. In this system, *N. lugens* females preferred to oviposit on plants that were infested with *C. suppressalis* compared to non-infested plants. Primary plant infestation by *C. suppressalis* leads to the production of a specific blend of HIPVs which are not attractive to *A. nilaparvatae*, so *N. lugens* escaped egg parasitism. HIPV blends can also induce counter-defences in pest insects, which subsequently perform better on plants emitting HIPVs compared to non-emitting plants [58]. Cultivars that are immune to the HIPV suppression of certain pests, that are inclusive of the natural enemies of all potential pest species, and that do not induce herbivore immunity are, therefore, important breeding targets.

Similar to herbivores, certain plant cultivars can be more attractive towards pollinators through the increased emission of the entire VOC blend or of specific compounds. For example, honeybees are inefficient pollinators of alfalfa, *Medicago sativa*, because the scent is not attractive, and therefore visitation rates are low [59,60]. Of the five floral compounds that do induce antennal responses in honeybees, only linalool, a common floral monoterpene, was found to be attractive [34,60]. Two compounds, 3-octanone and methyl salicylate, were repellent, and two compounds, (*Z)*-3-hexenyl acetate and ocimene, were neither repellent nor attractive. Developing cultivars with increased emission of linalool may increase pollination efficiency and thereby yields for alfalfa. Recent years have seen an increased effort in investigating floral scent composition and the compounds important for attracting pollinators in crops such as pear (*Pyrus* spp.) [61,62], kiwi (*Actinidia* spp.) [63], *Citrus* spp. [64], carrot, radish, and Chinese cabbage [65]. This will allow breeders to create cultivars that are more attractive to pollinators. Detailed knowledge of the multifunctionality of semiochemicals should allow for the development of cultivars with complex blends that are highly attractive for various pollinating insects and natural enemies of pests while repelling damaging pests.

### 2.4. Insect Pheromone Perception by Plants

Whilst semiochemicals in plant-insect interactions are typically considered in the context of insect responses to plant semiochemicals, insect-derived semiochemicals can also be perceived by plants and can activate plant defence pathways. The most studied elicitors are herbivore oral secretions (reviewed in [66]), but other semiochemicals, such as insect pheromones, can also be perceived by plants. Several studies demonstrate that tall goldenrod plants, *Solidago altissima*, can perceive the goldenrod gall fly sex-pheromone and enhance their defence responses to subsequent herbivory [67,68,69]. Furthermore, cotton plants can detect the aggregation pheromone of the boll weevil and activate indirect defence mechanisms attracting the parasitic wasp *Bracon vulgaris* [70]. Hence, the development of cultivars that eavesdrop on the intraspecific communication of their insect pests and strengthen their defences can offer a sustainable approach to reducing pest damage.

A deeper understanding of the role of semiochemicals for both plants and their associated insects will ultimately lead to innovative and sustainable agricultural practices. However, in the field, plants and insects are not the only players. A diverse community of microbes interacts directly with plants and insects, which can influence plant-insect interactions. These interactions are also mediated by semiochemicals and can further reveal methods to improve agricultural sustainability.

## 3. Insect-Microbe Signals

In addition to plant semiochemicals, microbial semiochemicals have also shown an important role in mediating insect behaviour, acting as cues for insect aggregation, suitable oviposition sites, or hosts (reviewed in [71,72]). These signals could be harnessed as attractants for improving the monitoring or trapping of pest insects. This topic will be discussed in the context of pre- and post-harvest pest management in this section.

### 3.1. Pre-Harvest Pest Management

Amongst the diversity of microbe-associated VOCs that insects encounter while foraging in pre-harvest environments, VOCs produced by yeast species mediate a range of microbe-insect interactions, especially the attraction of frugivorous insects. Attraction to yeast VOCs has been shown for the spotted wing drosophila, *Drosophila suzukii*, an invasive pest of ripening fruits (reviewed in [73]). This includes attraction towards a range of food-associated yeasts [74] and yeasts grown on different fruit species at different ripening stages [75]. Yeast VOCs can vary in strength and direction of attraction. For example, VOCs produced by certain yeast species differentially affect larval attraction and feeding in the cotton leafworm *Spodoptera littoralis* [75]. Interestingly, the attraction of *D. melanogaster* by yeast VOCs seems to be conserved across a phylogenetically broad range of yeasts, including yeast species which may pre-date the emergence of flowering plants [76]. Yeast-insect communication may, in fact, have contributed to the evolution of insect-mediated flower pollination due to the same attractant signals being detected in yeast and floral VOCs [75,76]. *Carpophilus* beetles, which are pests of ripening stone fruits in southern Australia, have also shown attraction towards gut-associated yeast VOCs [77]. The attraction of insects towards yeast VOCs has been demonstrated under open field conditions, highlighting the promise of these signals to be used in integrated pest management strategies. Traps containing *Metschnikowia pulcherrima* and *Hanseniaspora uvarum* captured *D. suzukii* under field conditions, although the combination of the two species did not significantly trap more insects than traps containing only *H. uvarum* [78]. Traps baited with the yeast-like *Aureobasidium pullulans* caught 1315 insects representing seven orders and 39 species, with 65% of the trapped insects being dipterans [79]. 2-Methyl-1-butanol, 3-methyl-1-butanol, and 2-phenylethanol were identified as major components of the headspace of *A. pullulans*. When synthetic blends of VOCs produced by yeast were incorporated into insect lures in olive orchards, greater numbers of the olive fruit fly *Bactrocera oleae* were captured compared to control traps, highlighting the potential for yeast VOCs to be incorporated into integrated pest management strategies [80]. Similarly, mated *B. tryoni* females were captured in greater numbers in lures containing a blend of fruit and yeast odours under field conditions [81]. This was also observed with the grapevine moth *Lobesia botrana*, where lures baited with 2-phenylethanol and acetic acid successfully trapped greater numbers of insects [82,83]. The greatest diversity of species oriented toward a 1:1 mixture of 2-phenylethanol and 2-methyl-1-butanol in spearmint plantations [79]. Thus, semiochemicals emitted by yeast can attract a range of insects, which offer alternatives to insecticides for horticultural pest management.

In addition to yeast-associated VOCs, VOCs from other microbes can also attract insect pests. Invasive ambrosia beetles in the *Euwallacea* nr. *fornicatus* species complex are established in California and Florida, USA. The Florida species was recently identified as *E. perbrevis* Schdl. which vectors fungal pathogens causing *Fusarium* dieback, a vascular disease that impacts avocado trees in southern Florida. *E. perbrevis* showed attraction towards VOCs produced by *Fusarium* sp. symbionts, with (*E*)-*p*-menth-2-en-1-ol being the primary attractant [84]. Bueno and colleagues (2020) have recently shown that a range of bacterial and fungal species isolated from *D. suzukii* and *D*. *suzukii*-infested fruits produce VOCs attractive towards the pest, including Proteobacteria and Actinobacteria species, as well as the yeast species previously discussed [74]. Bacteria isolated from the genitals of the phytophagous pests *Cyclocephala lunulata* and *C. barrerai* were attractive to the insects [85]. In addition to being attractive towards insect pests, microbial VOCs can attract natural enemies of insect pests. For example, natural enemies of aphids were attracted by bacterial VOCs [86,87,88], and insects across a range of trophic levels (prey, parasitoid and hyperparasitoid) were also shown to elicit olfactory responses to bacterial VOCs, the responses of which varied between and within trophic levels [89,90]. Findings from Goelen et al. (2020) were translated under greenhouse conditions, indicating that the results can be scaled up to agriculturally relevant conditions [88]. *Trissolcus basalis* parasitoid wasps, the main biological control agent of the stink bug *Nezara viridula*, were also attracted to certain bacterial strains which colonise the nectar of buckwheat, highlighting a role for nectar bacteria in the interaction between flowering plants and parasitoids [91]. Together, these studies highlight the potential for microbial VOCs to promote the conservation of biological control of insect pests. As well as their potential applications in monitoring and trapping insect pests of agricultural food crops, microbial VOCs could also be used for monitoring forest pests. One such example is the spruce beetle *Ips typographus*, a destructive forest pest of Norway spruce. Fungal symbionts of the beetle produce VOCs attractive towards *I. typographus* [92,93], which could be used to optimise semiochemical-based lures to monitor the spread of the beetle. Taken together, semiochemicals of microbial origin can be used to monitor and trap pest insects in a sustainable manner in crop and forest farming systems.

### 3.2. Post-Harvest Pest Management

Microbe-produced semiochemicals can be used to optimize post-harvest pest management [94]. Microbial signals are hypothesized to be important for stored product insects. At least one of the lineages of these insects likely evolved as animal cache exploiters. Many animals forget their caches, and in temperate environments, these are expected to mould, which would then be a reliable signal of their presence [94]. A recent systematic review found five stored-product arthropod species to be attracted by microbial VOCs from different sources, while four species were repelled [95]. Thirteen pests were unaffected or exhibited mixed effects towards microbial VOCs from different sources. The cosmopolitan, primary stored product insect, the lesser grain borer *Rhyzopertha dominica*, was attracted to wheat with moderate fungal damage, but a secondary pest, the red flour beetle, *Tribolium castaneum*, was not [96]. Notably, *R. dominica* bores into whole kernels and can be found large distances from food facilities on native tallgrass prairie in Kansas [97]. Beetles will even use acorns and other non-agricultural seeds as an alternate food source [98]. Thus, this species might have evolved on animal caches of food but is also attracted to suitable agricultural products that share the same or similar microbial cues. Recent work started to investigate the spatial scale of the attractiveness of microbial VOCs for stored product insects. The cosmopolitan rice weevil *Sitophilus oryzae* was attracted by microbial VOCs of wheat inoculated with *Aspergillus flavus* only at close range [95], while the cigarette beetle *Lasioderma serricorne* was attracted to fungal-inoculated wheat at both close and long-range (Ponce et al. in press). Microbial cues are a source of unique and highly attractive signals to stored product insects in food facilities environments that could be used to enhance the development of attract-and-kill programs in stored products (e.g., [99,100]).

## 4. Plant-Microbe Signals

Plants live in association with a diverse community of microbes in the soil, which can influence plant health through VOC production [101,102,103,104]. Microbes produce a range of structurally diverse VOCs across several chemical classes, including alcohols, ketones, hydrocarbons, aromatic compounds, terpenes, as well as sulfur and nitrogen-containing compounds (reviewed in [105]). These VOCs possess a range of biological activities that can influence plant development through plant growth promotion [106], direct inhibition of soilborne plant pathogenic fungi (reviewed in [107]), and the induction of defence responses against plant pathogens [108]. This section of the review will focus on the role of soil-borne microbial VOCs in induced plant defence against pathogens.

### 4.1. Plant Defence Induction by VOCs of Soil-Borne Microbes against Plant Pathogens

The VOC-mediated induced defence response was first demonstrated by Ryu and colleagues (2004), who showed that VOCs produced by *Bacillus amyloliquefaciens* induced resistance of *Arabidopsis thaliana* against the soft-rot pathogen *Erwinia carotovora*, for which 2,3-butanediol was involved in inducing resistance [108]. Since this work, 2,3-butanediol has been shown to induce plant resistance against a range of pathogens, including bacteria [109], viruses [110] and oomycetes [111]. Interestingly, defence induction by 2,3-butanediol appears to be enantiomer dependent, whereby (2*S*,3*S*)-butanediol was ineffective at inducing resistance against *E. carotovora* [112] and cucumber and tobacco mosaic virus [110]. As well as 2,3-butanediol, several other bacterial VOCs have shown a role in defence induction against bacterial pathogens, including 3-pentanol [113,114] and tridecane, produced by *Paenibacillus polymyxa* [115]. More recent work has also shown a role for 2-nonanone, produced by *B. velezensis*, in the activation of tomato defence against *P. syringae* under greenhouse conditions [116]. Whilst comparatively less studied compared to bacterial VOCs, fungal VOCs have also shown a role in activating plant defence against plant pathogens, including the *Trichoderma* VOC 6-pentyl-α-pyrone against *Alternaria brassicicola*, *Botrytis cinerea* and *Sclerotinia sclerotiorum* [117,118,119], *Cladosporium* VOCs against *P. syringae* [120] and *Talaromyces wortmanii* VOCs against *Colletotrichum higginsianum* [121]. Interestingly, VOCs produced by archaea can also induce plant resistance against *Pectobacterium carotovorum* and *P. syringae*, highlighting a relatively untapped source of defence-inducing VOCs which should be further explored [122]. Certain compounds also induced defence responses against *X. axonopodis* in pepper (3-penatnol) [114] and *P. syringae* in cucumber (3-pentanol and 2-butanone) [123] under open-field conditions. Together, these studies indicate that soil microbial VOCs demonstrate promise as alternative methods for crop protection through the induction of plant defences against a range of plant pathogens.

### 4.2. Microbe-Induced Plant Volatiles

In addition to directly influencing plant resistance, microbes can induce the production of plant VOCs (microbe-induced plant volatiles, MIPVs). MIPVs can directly inhibit pathogen growth, as well as induce plant resistance to pathogens (Reviewed in [124]). Recent examples include the VOCs 1-octen-3-ol, 3-octanone and 3-octanol produced by lima beans (*Phaseolus lunatus*), which were induced by *P. syringae* pv. tomato and dependent on the presence of the *P. syringae* type III effector HopP1 [125]. Specifically, synthetic 1-octen-3-ol activated a defence response against *P. syringae*. MIPVs elicited by beneficial soil microbes can also influence the rhizosphere microbiota of neighbouring plants through aerial semiochemical signalling. *Bacillus amyloliqufaciens*-inoculated tomato plants, *Solanum lycopersicum*, showed increased production of (*E*)-β-caryophyllene, which elicited salicylic acid production by neighbouring plant roots [126]. This caused synchronisation in the microbiomes of neighbouring and emitting plants (~69% similarity in microbial communities). Together, these findings indicate that soil microbes (beneficial or pathogenic) can influence the semiochemical signalling of plants, which can influence soil microbial communities. Recent work demonstrates that plant VOC changes, as a result of mechanical plant damage, can confer increased resistance to neighbouring plants against pathogens. Barley roots that were mechanically damaged produced a blend of VOCs, and when this VOC blend was exposed to undamaged receiver plants, increased resistance of receiver plants against the powdery mildew fungus *Blumeria hordei* was observed [127]. A holistic understanding of the semiochemicals emitted and received across these systems will help integrate these ecological processes for integrated pest management.

Plant VOC emission of (*E*)-2-hexenal can affect the interaction between host and pathogen pre-infection. In *B. cinerea*, constitutive levels of (*E*)-2-hexenal upregulate rate-limiting genes in the sulfur assimilation pathway of the fungus [128]. This process primes *B. cinerea* to utilize plant sulfate as a mechanism to mitigate oxidative stress, facilitating plant infection. As sulfate assimilation is conserved across the fungal kingdom [129], pre-infection priming of fungal pathogens through constitutive plant VOCs may be common. In contrast to the priming effects of plant VOCs on fungal pathogens, pathogens can also manipulate host VOC profiles to reduce emission and VOC-induced resistance in neighbouring plants. During infection by *Sclerotinia sclerotiorum* on potato, *Solanum tuberosum*, the pathogen downregulated biosynthetic genes for VOC precursors, which led to no change in plant VOCs profile quality or quantity post-infection and subsequently no induced resistance in neighbouring plants [130].

The importance of microbial VOCs and MIPVs on plant development demonstrates promise for their use as crop protection tools. Investigating the role of these VOCs on plants under glasshouse/field conditions is an important priority for future research, as well as the impact of these VOCs on the soil microbiome. Another important area of research for gaining a holistic understanding of plant-microbe VOC signalling is the impact of these signals on insects as a tripartite system. An overview of the semiochemicals involved in plant-associated insects and microbes is illustrated in Figure 2.

## 5. Plant-Microbe-Insect Signals

The effects of microbes on plant-insect interactions can be triggered by microbe colonisation and the associated changes in plant semiochemical production or plant responses to microbial VOCs, including changes in plant or insect semiochemical production. In this section, we will discuss how microbes influence plant interactions with herbivores and their natural enemies, pollinating insects, as well as how insect-associated microbes influence plants.

### 5.1. Influence of Plant-Microbe Interactions on Herbivores 

Plant-microbe-insect interactions can trigger plant immune responses to biotic stressors, thereby modulating insect behaviour as well as microbe pathogenesis [131]. It is well-documented that plant-associated microbes can affect plant quality and defence, triggering the production of semiochemicals that provide both direct and indirect protection against herbivores [132]. One such example is the improved resistance against the tobacco peach aphid when sweet pepper plants are inoculated with the entomopathogenic fungus *Akanthomyces muscarius*. Aphids showed increased attraction towards inoculated plants compared to non-inoculated plants, as well as reduced longevity and fecundity when feeding on *A. muscarius* inoculated plants [133]. Similarly, endophytic colonization of melon plants by entomopathogenic fungi increases *Aphis gossypii* mortality [134]. Pathogen-infected plants can negatively affect herbivore life history, causing decreased larval development, performance and increased mortality [135,136]. The inoculation of plants with beneficial bacteria, such as plant growth-promoting rhizobacteria (PGPR), has been shown to confer herbivory protection and/or induce VOC production. One of the pioneering studies demonstrating that PGPR can confer protection against insects was performed using *Bacillus pumilis* strain INR-7. Since then, several examples have been reported demonstrating that rhizobacteria species and strains negatively affect herbivore development and performance [137,138,139,140,141,142,143]. However, Pineda et al. (2012) showed that treating *A. thaliana* with *Pseudomonas fluorescens* WXS417r positively affected the weight gain of the generalist aphid *Myzus persicae*, while no effect was detected on the crucifer specialist aphid *Brevicoryne brassicae*, demonstrating that rhizobacteria effect markedly differs upon herbivores from different dietary specialization [144]. In addition to direct microbial colonisation, plant responses to microbial VOCs can influence plant resistance to herbivores. VOCs from the soil-borne fungus *F. oxysporum* induced phenotypic responses of *Brassica rapa*, which differentially affected the performance of the root herbivores *Heterodera schachtii* and the cabbage root fly *D. radicum* [145]. The identity of the source of microbial VOCs is important to determine the effects on the outcome of plant-insect interactions. VOCs from 11 different pathogenic and non-pathogenic soil-borne fungi increased the susceptibility of *A. thaliana* to the generalist herbivore *Mamestra brassicae* [146]. The effect of different fungi on plant susceptibility did not depend on pathogenicity but was fungal species specific. Taken together, microbial colonisation and VOCs can influence plant resistance to insect pests that can potentially be utilised by sustainable agriculture.

### 5.2. Influence of Plant-Microbe Interactions on Natural Enemies’ Behaviour

Microbes can alter plant indirect defence responses, i.e., through the recruitment of herbivore antagonists, such as predators and parasitoids. Guerrieri and colleagues (2004) first demonstrated that tomato plants inoculated with the arbuscular mycorrhizal fungus *Glomus mosseae* significantly increase the attraction of the parasitic wasp *Aphidius ervi* towards undamaged plants [147]. A similar phenomenon has been observed with the predator *Macrolophus pygmaeus*, which is attracted to VOCs emitted by uninfested tomato plants inoculated with the soil fungus *Trichoderma longibrachiatum* [148]. By contrast, the aphid parasitoid *Diaeretiella rapae* was less attracted towards VOCs produced by aphid-infested *A. thaliana* colonized by *P. fluorescens* [144], showing that microbes can modify herbivore-induced plant VOCs and impair natural enemy attraction. Several studies have demonstrated that the treatment of plants with PGPR can also alter the emission of plant VOCs, with important ramifications for plant-insect and tri-trophic interactions [140,142,144,149]. For example, Ngumbi (2011) demonstrated that PGPR-treated cotton plants produce a distinct blend that is qualitatively different from untreated cotton plants [149]. The blend produced by PGPR-treated plants was shown to be repulsive to ovipositing *S. exigua* while highly attractive to parasitoids [149,150]. Similarly, Pulido et al. (2019) reported quantitative differences in the VOC profiles of soybean plants treated with PGPR and nodule-forming beneficial rhizobacteria, enhancing the attraction of the parasitoid wasp *Pediobious foveolatus* [151]. These examples demonstrate the influence soil microbes can have on plant VOC production and the subsequent alteration of parasitoid behaviour, which could be harnessed for the biological control of herbivores.

### 5.3. Influence of Plant-Microbe Interactions on Pollinator Behaviour

In addition to producing semiochemicals involved in plant defence against herbivores, microbes can alter signals used by flowering plants to attract flower visitors. Plant growth-promoting effects by beneficial microbes potentially enhance floral VOC signals and thereby increase pollinator visitation and plant reproduction. Indeed, although arbuscular mycorrhizal fungi (AMF) have been shown to enhance pollinator visitation and plant seed production, the role of semiochemicals is poorly explored [152], and similarly for PGPR (see [153]). In the only study on this topic to our knowledge, Becklin and colleagues (2011) showed that AMF may reduce floral scent emission. This did not affect pollen receipt but reduced flower damage by ants [154]. Opposite to beneficial microbes, pathogenic microbes can stunt plant growth and thereby potentially reduce floral VOC signals and pollinator attraction. Interestingly, certain plant pathogens can alter floral VOC emission and actually enhance pollinator attraction. For example, cucumber mosaic virus infection alters the floral scent emission of tomato (*Solanum lycopersicum*) plants, which was shown to increase the attraction of *Bombus terrestris* bumblebees [155]. After choosing a plant to visit, pollinators interact with individual flowers. Moreover, on this small scale, microbes can influence pollinator visitation by influencing VOC production. Floral nectar provides a habitat for a range of microorganisms. Growing bodies of evidence highlight the importance of nectar microbes in semiochemical signalling between plants and nectar-feeding insects [156]. For example, the presence of nectar-inhabiting yeasts and bacteria can modify nectar chemistry and result in different VOC blends that modulate the olfactory responses of the aphid parasitoid *Aphidius ervi* [157], the bumblebee *B. terrestris* [158,159] and the honeybee *A. mellifera* [160]. Nectar microbe-induced changes in floral scent result from compounds emitted by the microbes when metabolizing the nectar or when the microbes metabolize VOCs produced by the plant [156,161]. Recently, the combination of bacteria and yeast was shown to cause additive changes in nectar scent, which can increase honeybee visits [162]. This indicates that microbes should be considered consortia rather than individual species when determining their effects on floral VOC changes. Taken together, microbes can have contrasting effects on pollinator attraction by influencing flower scent. Inoculating crops with specific microbes to manipulate pollinator attraction can be an interesting tool for sustainable agriculture.

### 5.4. Influence of Insect-Associated Microbes on Plant VOC Emissions

In addition to plant-associated microbes, insect-associated microbes can also modify plant-insect interactions, specifically plant defences, in response to herbivore attacks. 

Microbes can manipulate their insect vector behaviour directly or by manipulating plant semiochemicals as a strategy to maximize their acquisition and transmission from plant to plant [163]. For example, the bean common mosaic necrosis virus (BCMNV) induces qualitative changes in the VOC blend of *P. vulgaris*, including suppression of the aphid attractant α-copaene. These changes are sufficient to influence aphids’ preference for uninfected plants over BCMNV-infected bean plants [164]. Similarly, VOCs emitted by tomato plants infected with the cucumber mosaic virus (CMV) attract the aphid vector *M. persicae* early on in the infection [165]. As the infection develops, however, aphids are repelled by CMV-infected plant VOCs and attracted to healthy ones. In addition to influencing plant VOC emission, plant-associated microbes can directly influence insects. Recent work shows that opportunistic fungi can, in fact, manipulate insect herbivore hosts to facilitate plant infection and promote dissemination. Franco and colleagues (2021) showed that a VOC blend emitted by the fungal phytopathogen *Fusarium verticillioides* is attractive to its host, the caterpillar *Diatraea saccharalis,* which contacts the fungus when feeding on infected plants [166]. Once *D. saccharalis* caterpillars become adults, the fungus is then vertically transmitted to their offspring, which inoculate the pathogen into healthy plants. Females carrying the fungus also prefer to lay eggs on healthy sugarcane plants, whilst non-contaminated females are attracted to fungus-infected plants. This tripartite pathosystem shows a complex manipulation of both host insect and plant via VOC profile changes. Within the current literature, it is clear that plant-associated microbes can influence plant-insect interactions via a range of semiochemicals. Moreover, the use of microbes might provide a viable alternative to chemical pesticides and has great potential to manage insect pests sustainably.

## 6. Application of Semiochemicals to Improve Sustainability in Agriculture

### 6.1. Examples of Pest Management Products Based on Semiochemicals

Several products based on semiochemicals produced by plants, insects and microbes have been commercialised, highlighting their potential to address agricultural challenges. For example, PredaLure (AgBio Inc., Westminster, CO, USA) is a commercially available semiochemical lure used for the attraction of natural enemies of agricultural pests based on the herbivore-induced plant VOC methyl salicylate. Traps baited with PredaLure trapped greater numbers of several natural enemies of insects, including hoverflies, lady beetles, and green lacewings, compared to unbaited traps, demonstrating the efficacy of the lures under agriculturally relevant conditions [167]. Insect pheromones are also widely used in pest management practices for monitoring, control by mass trapping, lure-and-kill, and mating disruption [168]. In terms of microbially produced semiochemicals, PALADIN™ is a pre-plant soil fumigant based on dimethyl disulfide (DMDS), which is produced by a range of microbial species, as well as species within the *Alliaceae* family [169] and can be used for the control of nematodes as well as soil fungal pathogens [169,170,171].

Laurel wilt, a vascular disease of trees in the family Lauraceae, has caused extensive mortality in native *Persea* species and avocado (*P. americana*), an economically important fruit crop [172,173,174]. Effective lures for early detection of the redbay ambrosia beetle, *Xyleborus glabratus* are critical to slow the spread of laurel wilt, although no pheromones are known for this species. Moreover, the beetle is not attracted to ethanol, which is the standard lure employed for the detection and monitoring of ambrosia beetles in the US [175]. Kendra and colleagues showed that a combination of copaene lures and the standard quercivorol lures (containing *p*-menth-2-en-1-ol isomers, volatiles from symbiotic fungi) resulted in synergistic captures of *E*. nr. *fornicatus* [176]. This combination lure has been adopted by SAGARPA (Mexico’s Secretaría de Agricultura, Ganadería, Desarrollo Rural, Pesca y Alimentación) in monitoring programs for both *E*. nr. *fornicatus* and *X. glabratus* in high-risk areas (Mexican ports, international borders, and avocado production regions) [177]. Together, these examples highlight that the commercialisation of compounds produced as semiochemicals between plants, insects, and microbes can be used as alternative crop protection tools to chemical inputs.

Multiple commercial products have also been developed to increase the foraging efficiency of pollinators and ultimately increase fruit yield or reduce cost by replacing hand pollination [178,179,180,181]. Many of these products consist of a synthetic mixture of chemical attractants, such as floral VOCs, plant sugars, and insect pheromones. The Nasonov pheromone, in particular, is utilised in these products to attract honeybees [182,183]. This pheromone is released by worker honeybees to orient other foragers to nectar sources and the hive [184]. The synthetic version of this pheromone consists of a 2:1 ratio of citral and geraniol but can also contain other chemical components [185]. The results of the usage of these products are mixed. For example, Jailyang et al. (2022) reported an increase in pollinator visitation when the commercial product Bee Scent was applied to kiwifruit [181]. This increase in visitation improved fruit set, yield, and higher grading of the fruit. In contrast, Williamson et al. (2018) tested two commercial products utilising natural attractants and the Nasonov pheromone [180]. No measurable increases in pollinator visitation or fruit set were detected in apples, blueberries, or cherries. Continued research is required to elucidate how best to formulate and maximise these products, as any enhancements to pollination would be extremely beneficial to agriculture.

### 6.2. Push-Pull

Push-Pull technology for integrated pest and weed management in crops is based on the understanding and application of chemical ecology. This technology has been implemented successfully in sub-Saharan Africa to protect maize and other crops from stem borers and *Striga* weeds. Push-pull incorporates companion plants that are grown in between and around the main food crop as a means of crop protection against insect pests and weeds. These companion plants emit semiochemicals that act either as a repellent (push) or attractant (pull) towards insect pests [186,187,188] (Figure 3). The semiochemicals released by the companion crops also make it possible to exploit natural populations of beneficial organisms such as parasitoids, which in turn reduce herbivore damage. One example of push-pull in action involves the legume *Desmodium* spp. (companion crops) as the “push” crop, and the perennial Napier grass, *Pennisetum purpureum*, as the “pull” crop, which can be planted in between food crops, including maize or sorghum. *D. uncinatum* is known to produce *C*-glycosylated flavonoids, di-*C*-glycosylflavones, which suppress the growth of *Striga* weeds, and (*E*)-ocimene and DMNT, which repel stemborer pests and recruit natural enemies [187]. Concurrently, the maize is surrounded by a border of Napier grass known to emit 4-allylanisole, octanal, nonanal, naphthalene, eugenol, and linalool that attract the stemborer pests away from the main crop [186,187]. This provides a cost-effective means of pest management since it relies on readily available indigenous plants rather than expensive external inputs and enhances maize yields while addressing considerable production restrictions. Together, the push-pull strategy reduces pest infestation and subsequent yield losses of the main crop without the need to apply insecticides.

As well as maize and sorghum, companion plants and push-pull approaches are being investigated in various other cropping systems. These include soybean, *Glycine max*, and its stink bug pests, tomato and a variety of pests (including whiteflies and tomato pinworms), various *Brassica* crops with pests including pollen beetles, and coffee, *Coffea arabica*, with its leafminer and berry borer pests [189,190]. Recent work has focussed on the deterrent/push/repellent/masking function of companion plants, especially under greenhouse conditions. For example, VOCs emitted by coriander were not repellent by themselves but reduced the attractiveness of tomato VOCs for the silverleaf whitefly, *Bemisia tabaci* [191]. VOCs from oregano were also found to be repellent and mask tomato VOCs for the silverleaf whitefly [192]. Potentially, the repellent compounds in the oregano VOC blends were (*E*)-β-caryophyllene and α-humulene, also proposed as repellent compounds for stem borer pests in the maize/sorghum system. Hence, coriander and oregano provide odour masking of tomato VOCs and/or directly repel whitefly pests, although the active semiochemicals were not identified. Intercropping rosemary, *Rosmarinus officinalis*, alongside sweet pepper, *Capsicum annuum*, significantly reduced populations of several major pests of sweet pepper, without reducing populations of natural enemies of the pests [193].

A conceptual framework for the use of intercropping and the use of companion plants for increased pollination of crops has recently been established, linking pollinator facilitation to ecological theory [194], highlighting a promising avenue for future research. Co-flowering plants such as *Trifolium repens* (clover) *Taraxacum officinale* (dandelion), and *Plantago lanceolata* (plantago) increased bee visitation to apple flowers, although the role of semiochemicals was not investigated [195]. Many plants share floral scent compounds, with limonene, (*E*)-ocimene, myrcene, linalool, α-pinene, and benzaldehyde present in the blend of more than 60% of all angiosperm families [34]. Intercropping and companion plants thereby harbour the great potential to attract pollinators from the surrounding landscape to the crop using general signals for the presence of flowers. Designing cost-effective push-pull systems that manage multiple pests while increasing the attraction of pollinators and natural enemies will contribute greatly to more sustainable agricultural practices.

### 6.3. Transgenic Approaches for Insect Pheromone Biosynthesis in Plants

Insect pheromones offer an environmentally sustainable alternative to synthetic pesticides, which can be integrated into insect pest management strategies. However, the costs to produce pheromones, and the instability of volatile pheromones, when applied seasonally to fields, pose problems for their deployment in agriculture [196]. Therefore, the possibility for plants to be engineered to synthesise insect pheromones is an alternative approach for sustainable pest management which is receiving increasing attention. Examples of this include *Arabidopsis* [197] and wheat [196], which have been successfully engineered to produce the aphid alarm pheromone (*E*)-β-farnesene, which is released by aphids to alert other aphids of natural enemies whilst also increasing the foraging of their natural enemies. However, when grown under open field conditions, transgenic wheat showed no reduction in aphid populations. This may have been due to wet weather or differences in release rates of the pheromone from aphids, which produce the compound in sudden bursts following an attack by a predator, versus the transgenic plants, which continuously produce the compound [196]. Nonetheless, the study was promising, demonstrating for the first time that food crops could also be engineered to produce insect pheromones. More recent work has shown that *Camelina sativa* seeds can be engineered to produce the insect sex pheromone precursor (*Z*)-11-hexadecenoic acid, showing plants can act as factories for pheromone production [198]. The pheromone precursor was isolated from transgenic plants and converted into the final pheromone, which was then added to traps under field conditions. Results demonstrated that plant-derived pheromone traps were equally as effective at catching the diamondback moth, *Plutella xylostella*, and cotton bollworm, *Helicoverpa armigera*, as traps containing the synthetic pheromone which was commercially produced. *Nicotiana benthamiana* has also been engineered to produce (*Z*)-11-hexadecenol, (*Z*)-11-hexadecenal and (*Z*)-11-hexadecenyl acetate, the major sex pheromone components of the rice stem borer (*Chilo suppressalis*) [199]. Therefore, the ability of plants to be transformed to produce these pheromones could be harnessed to reduce the costs of pheromone production and improve the stability of pheromone release under agricultural conditions [200]. Similarly, the yeast species *Yarrowia lipolytica* has been engineered to produce insect pheromones from species including *Helicovera arigera* (cotton bollworm) and *Spodoptera fruiperda* (fall armyworm), demonstrating the potential for microbes to act as factories for pheromone production [201].

## 7. Future Perspectives and Research Priorities

In summary, these studies highlight the potential for plant, insect and microbial semiochemicals to be harnessed as crop protection tools for sustainable agriculture. Several key research priority areas are highlighted below, which should be the focus of future work to advance our knowledge of plant-microbe-insect semiochemicals and their use in sustainable agriculture.

Field testing of bioactive semiochemicals. Laboratory-based findings which occur under controlled conditions should be scaled up to more ecologically relevant conditions through greenhouse and field trials. This includes determining the most appropriate methods for administering semiochemicals (e.g., through slow-release formulations and dispensers) that mimic their biological origins. These areas of research can determine the feasibility of using semiochemicals under open-field conditions and underpin their commercialisation. Moreover, whilst many studies focus on the bioactivity of individual compounds, it may be that compounds produced as blends by emitting organisms provide a specificity that may elicit different behavioural responses by the receiving organisms, which warrants further study.

Use of VOCs for pathogen/pest detection. VOCs produced by plants could provide non-invasive methods to detect pests prior to the onset of physical symptoms. MIPVs have recently been exploited to detect a range of bacterial pathogens in tomatoes under open field conditions through changes in characteristic leaf VOC emissions following pathogen inoculation [202]. Similarly, the use of an electronic nose (e-nose) was also able to discriminate between healthy, mechanically damaged, whitefly-infested and aphid-infested tomato plants under greenhouse conditions [203,204]. In addition, z-nose (an ultrafast portable GC analyzer) has the potential to detect VOC signatures diagnostic of fruit fly (Tephritidae) infestation in citrus [205]. Together, the VOC changes occurring as a result of pathogen or pest infestation are a potential tool for more rapid detection.

Understanding the molecular basis of semiochemical signal production/perception. Understanding the genetics underpinning the biosynthesis of semiochemicals (plant, insect, or microbial) could enable the engineering of plants to enhance the production of beneficial semiochemicals. Gene editing tools, including CRISPR/Cas9, could then be used to manipulate plant defence mechanisms, including the production of induced VOCs for the recruitment of natural enemies and microbes for the production of plant defence eliciting compounds. A greater understanding of how semiochemicals are perceived by receiving organisms is also required.

Understanding the impacts of climate change on semiochemical signalling. Abiotic stressors are known to affect plant metabolism and VOC production [206], which can influence defensive traits against herbivores and signal perception for natural enemies and pollinators [207,208]. A key challenge may be maintaining the efficacy of lures in the face of exacerbating climate change, with certain studies suggesting quicker depletion of lures and lower efficacy under warmer regimes in the field [209]. Warmer temperatures could also produce shifts in insect responses to certain semiochemicals, thereby hampering the effectiveness of chemical communication systems as well as attempts to exploit them (reviewed in [210]). A better and more widespread understanding of how climate change is affecting semiochemical signalling among insects in order to deliver reliable solutions is needed. Solutions may be more easily attainable if related to the depletion of lures given new matrix technology for slow-release formulations [94] and technology on the horizon for creating complex blends timed correctly and in just the right amounts over extended periods.

Improved high-throughput analysis of chemical ecology datasets. Increasingly, research projects are investigating whole volatilomes or metabolomes, which have historically required a lot of data processing power and time investment to process the data. Streamlining this analysis with new data science tools and machine-learning algorithms will greatly speed the translation of new data into new discoveries. One such new R package includes *uafR* [211], which can reduce data processing times from weeks to minutes. Packages like this and others will be able to speed discovery in the chemical ecology space. Moreover, the use of data repositories for data sharing across wider academic communities should be adopted.

Protection of natural areas. Significant research has been conducted to investigate how plant-insect-microbe interactions can be manipulated to improve agricultural sustainability. However, the principles of chemical ecology have not been readily adopted to protect non-crop areas, which are usually dominated by native vegetation and provide ecosystem services critical to maintaining the productivity of many agricultural systems, or grasslands which are used for pasture which can be damaged by pests and pathogens [212]. While the enhancement of biological control has been achieved in agricultural crops, more research will be needed to determine if these methods are suitable to aid land managers in protecting natural areas from invasive and/or exotic plants and insects. One example of research incorporating chemical ecology into the management of natural areas is the investigation of herbivore-induced plant VOCs to protect areas of *Phragmites australis* from the invasive scale insect *Nipponaclerda biwakoensis* in the Mississippi River Delta [213]. Differential responses of specific lineages of *P. australis* to the scale insect may result in enhanced or reduced biocontrol services from the parasitoids of the scale insect. It is believed by researchers that a better understanding of the compounds governing the plant-scale insect-parasitoid interactions could result in better mitigation programs for the invasive scale insect, especially when informing restoration efforts.

Biological control of weeds. In addition to the biological control of insects and pathogens, increasing attention is being paid to how chemical ecology can be applied to enhance classical weed biological control [214,215]. This involves the introduction of highly host-specific coevolved herbivores to provide permanent suppression of invasive weeds. Many of the concepts of chemical ecology are recently being applied to improve the predictability and safety of weed biological control agents, including the compounds which govern the underlying host specificity of these agents, how the chemistry of the invasive weed can evolve after introduction, how herbivore-induced compounds will impact the biological control agent, intraspecific variation of secondary plant compounds, and sequestration of defensive compounds by the biological control agents [214]. For example, the incorporation of the olfactory cues governing the host-specificity of the potential biological control agent *Mogulones borraginis* and its host plant allowed for this insect to be petitioned and recommended for release in the United States [216,217]. Beyond its importance to the predictability and safety of weed biological control programs, chemical ecology can impact weed biological control programs by increasing monitoring efficacy [218,219], increasing establishment [220,221], and enhancing the damage potential of the agents [222,223]. The use of semiochemicals to aid in the monitoring of the gorse pod moth, *Cydia succedana,* allowed land managers to determine the minimum amounts of the insects to get establishment, eliminating the need for costly mass releases. Pheromones and plant VOCs have also been utilised to monitor the northern tamarisk beetle, *Diorhabda carinulata,* allowing land managers to detect populations in areas not previously released. Early detection of biological control agents can allow managers to incorporate the biocontrol program into their pest management strategies, providing new strategies for land managers to address invasive pests in natural areas and allowing for the protection of the ecosystem services these areas provide to agriculture.

Improved detection and monitoring of invasive insect pests. Invasive insect pests pose a major threat to natural ecosystems and agricultural lands. New invasive insect pests can cross overseas, dominate quickly over extended areas and cause severe damage to new ecosystems. Once an alien species establishes a new habitat, eradication of the species is difficult, and controlling them can be costly. Research should prioritise understanding the ecology of invasive pests and aim to identify host locations and feeding cues that can be formulated for effective use in monitoring and detection programs for specific invasive insect species. Early detection tools based on semiochemicals, such as monitoring, mass trapping, lure-and-kill, mating disruption, and push-pull strategies, are needed to suppress or eliminate alien species in new regions. Additionally, remote sensing and imaging technologies will provide rapid early detection of exotic insect pests. Chemical ecology research can be performed using multidisciplinary approaches, combining laboratory assays, electrophysiology experiments, chemical analyses, and field trials, which are needed to develop effective strategy methods for the management of introduced alien invasive insect species. This can improve the detection, monitoring, control, and eradication of new invasive insect species.

Sampling semiochemicals under more ecologically relevant conditions. The retrieval of all relevant semiochemicals from plant root exudates and soil is often performed under conditions that are different from those encountered in field-grown plants, making the extrapolation of findings challenging. Consequently, there is a need for research on analytical methods that can assess root and microbe-secreted chemicals while preserving their quality, quantity, and temporal dynamics without disrupting microbial viability. One solution is a microdialysis-based analytical system involving probes positioned in soil microsites that continuously monitor and detect dynamic changes in root-secreted chemicals within the plant-soil system [224,225]. A comprehensive investigation of plant semiochemicals is necessary to comprehend their interactions with other biotic entities, such as plant competitors, herbivores, and pathogens.

## Figures and Tables

**Figure 1 biomolecules-13-00997-f001:**
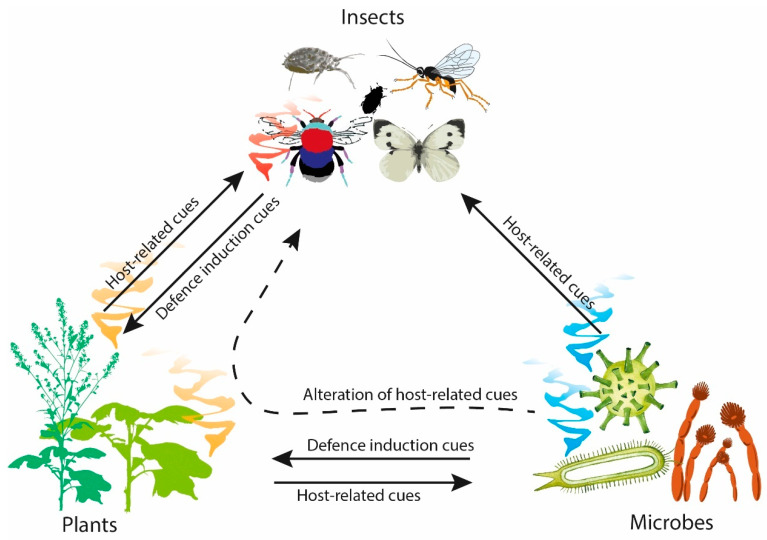
Overview of the semiochemical interactions between plants, insects and microbes.

**Figure 2 biomolecules-13-00997-f002:**
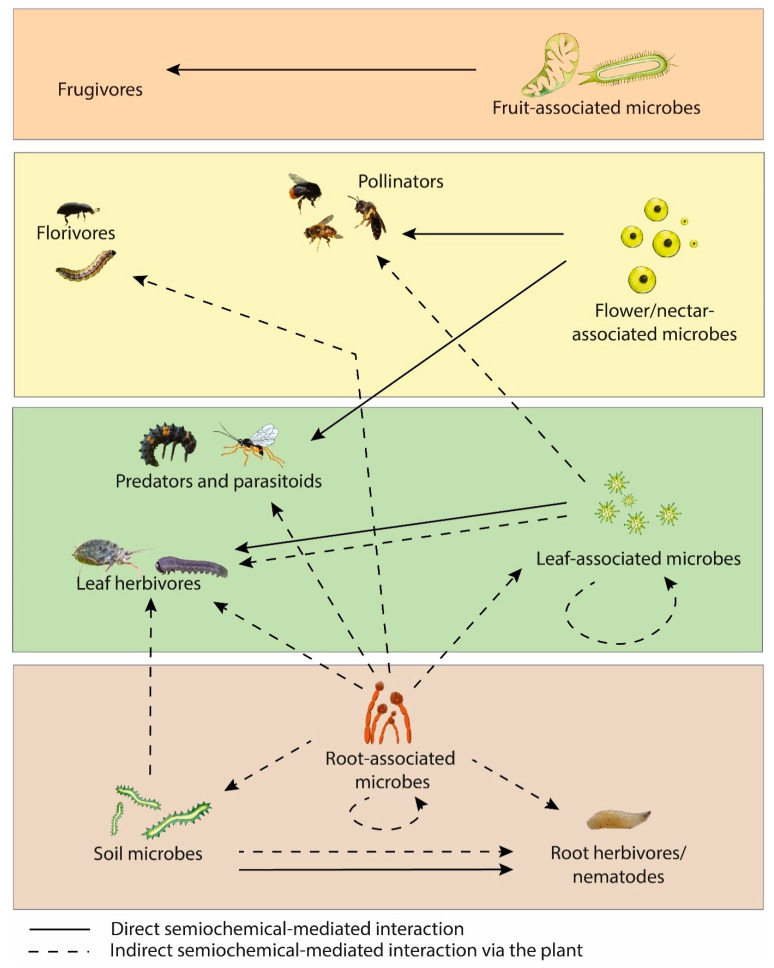
Semiochemical interactions between plant-associated insects and microbes.

**Figure 3 biomolecules-13-00997-f003:**
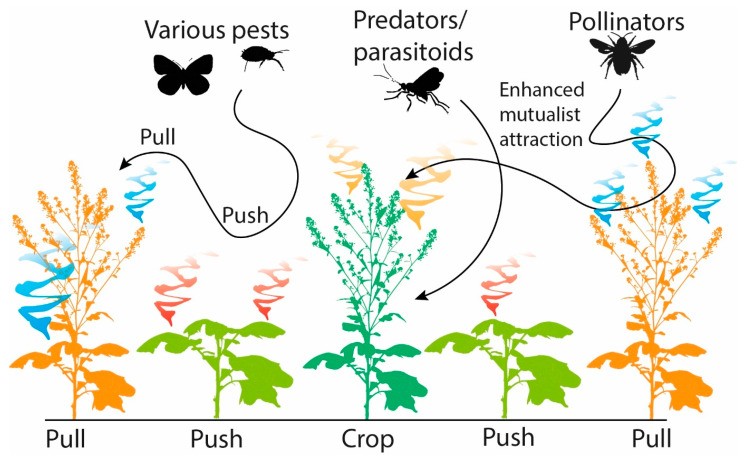
The role of semiochemicals in push-pull systems.

## Data Availability

The data supporting reported results can be found in the manuscript.

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
