# Peer review of "Deciphering Plant-Insect-Microorganism Signals for Sustainable Crop Production"

_biomolecules, 2023, doi:10.3390/biom13060997_

Round 1

Reviewer 1 Report

The submitted manuscript well organized and introduced recent study of chemical ecology. It is worthy to be published in Biomolecules. 

Before acceptance, there are some minor point to be amend. 

The references did not describe the volume number and pages. Please indicate it. Some books are listed as like Journal. Please follow the citation rule for Book. 

Author Response

The submitted manuscript well organized and introduced recent study of chemical ecology. It is worthy to be published in Biomolecules. 

Before acceptance, there are some minor point to be amend. 

Response: The authors would like to thank the reviewer for their positive feedback on the manuscript.

Response: We have been through the attached file and made the amendments you suggested.

The references did not describe the volume number and pages. Please indicate it. Some books are listed as like Journal. Please follow the citation rule for Book. 

Response: The references have now been updated.

Reviewer 2 Report

Thomas et al.

Deciphering plant-insect-…

General

Impressive review, very comprehensive, admirable effort. Certainly publishable.

General comment

Since abstract/intro set the applicationcrop protection//food security theme, one would ideally expect that an effort is made to distinguish between "facts and phantasies", i.e. research aiming at future applications and research that has led to or is connected to existing practical applications.

This is, admittedly, not an easy task, since the focus of scientific literature is, of course, on current research, while reports of current real-life applications and their coverage are rare.

However, attempts should nonetheless be made to delineate between areas/fields of research where future applications are only expected/proposed, and research that has already given rise to existing applications.

This concerns, most chapters, and especially 2.2, 2.3, 2.4

531. Another example. Selection criteria for research cited here remain unclear. This is fairly striking, since there are so many examples for efficent/widespread applications. Instead, some of the species/methods mentioned here are probably not very widely used, if they have left the idea stage at all. 

This particular area, holding material for a very large review article all by itself, also shows that it may be difficult or even impossible to adequately review the entire research field.

One way of dealing with this would be to clearly state that only few, selected examples are given. Ideally, one would even identify/define the criteria for selecting case studies.

Specific

37. pose … "issues" (?)

41. plants, microbes and animals

89. move figure one line down

124. reference(s)?

233. "innovative and sustainable … practices" have been mentioned before, already, but the interconnection with this section (2.4) is quite elusive.

Author Response

General

Impressive review, very comprehensive, admirable effort. Certainly publishable.

Response: The authors would like to thank the reviewer for their positive comments.

General comment

Since abstract/intro set the applicationcrop protection//food security theme, one would ideally expect that an effort is made to distinguish between "facts and phantasies", i.e. research aiming at future applications and research that has led to or is connected to existing practical applications.

This is, admittedly, not an easy task, since the focus of scientific literature is, of course, on current research, while reports of current real-life applications and their coverage are rare.

However, attempts should nonetheless be made to delineate between areas/fields of research where future applications are only expected/proposed, and research that has already given rise to existing applications.

This concerns, most chapters, and especially 2.2, 2.3, 2.4

Response: The authors thank the reviewer for their comment. The aim of our review is to give an overview of how ecological interactions between plants, insects, and microbes driven by semiochemicals can potentially be manipulated to increase agricultural sustainablility. We do not mean to discuss the process of translating research to application, although we do discuss this in section 6 and 7. Although this is a very interesting topic, we feel it falls outside the scope of our review, but could form an interesting, separate review in itself.

531. Another example. Selection criteria for research cited here remain unclear. This is fairly striking, since there are so many examples for efficent/widespread applications. Instead, some of the species/methods mentioned here are probably not very widely used, if they have left the idea stage at all. 

This particular area, holding material for a very large review article all by itself, also shows that it may be difficult or even impossible to adequately review the entire research field.

One way of dealing with this would be to clearly state that only few, selected examples are given. Ideally, one would even identify/define the criteria for selecting case studies.

Response: This is a great point, which we have hopefully addressed at the end of the introduction. We have included a sentence justifying the literature we have selected: The examples we have selected are based on more recent studies, whilst also including earlier examples where certain phenomena were first established, to provide broader context.

Specific

37. pose … "issues" (?)

41. plants, microbes and animals

89. move figure one line down

124. reference(s)?

233. "innovative and sustainable … practices" have been mentioned before, already, but the interconnection with this section (2.4) is quite elusive.

Response: The authors have made the suggested amendments to the article

Reviewer 3 Report

The authors have set themselves a very ambitious target in reviewing “Plant-Insect-Microorganism Signals”. They have made a good attempt at this, and I think the main strength of the review is to highlight the multi-faceted nature and complexity of these interactions which are often considered in more narrow subsets.

Such a wide-ranging review is inevitably selective in places, and my only serious criticism would be that this is too selective in some places. This is particularly true in Section 6 on application of semiochemicals.

·       Thus selection of BMSB and emerald ash borer as examples of use of pheromones is hardly representative as, in fact, the synthetic pheromones of these two species, particularly the latter, are not as attractive as in many other species and non-chemical factors are at least as important.  I think the authors just have to say that pheromones are now widely use in pest management for monitoring and control by mass trapping, lure-and-kill and mating disruption and give a reference such as Lucchi A, Benelli G  (2022) From Insect Pheromones to Mating Disruption: Theory and Practice. https://doi.org/10.3390/books978-3-0365-3178-6

·       The section on redbay ambrosia beetle seems unjustifiably long at nearly a page, perhaps driven by the interest of one of the contributors, and could be greatly shortened to allow inclusion of other examples.

·       Semiochemicals produced by yeasts are highlighted earlier and so it would be good to mention widespread use of microbial volatiles as lures for a range of insects, particularly fruit flies such as SWD.

·       In Section 6.3. it could be mentioned that one of the most significant and commercially-advanced uses of transgenic approaches is that of BioPhero using transgenic yeasts to produce pheromones on large scale commercially. Holkenbrink et al. (2020) https://doi.org/10.1016/j.ymben.2020.10.001

Other relatively minor editorial comments are as follows.

L 171, 379, 621. (E)-β-caryophyllene

L209 and throughout. Standardise on E and Z rather than cis and trans in places

L451. “Behaviour of natural enemies” is less clunky

L479. the role of semiochemicals has been poorly explored [152], and similarly for PGPR (see [153]).

L514. As the infection develops, however, aphids are re-

L530. 6. Application of semiochemicals to improve sustainability in agriculture

L534. Predalure is a semiochemical lure, not pheromone lure

References. Insect names should be in italics?

Author Response

The authors have set themselves a very ambitious target in reviewing “Plant-Insect-Microorganism Signals”. They have made a good attempt at this, and I think the main strength of the review is to highlight the multi-faceted nature and complexity of these interactions which are often considered in more narrow subsets.

Response: The authors would like to thank the reviewer for their positive comments.

Such a wide-ranging review is inevitably selective in places, and my only serious criticism would be that this is too selective in some places. This is particularly true in Section 6 on application of semiochemicals.

Thus selection of BMSB and emerald ash borer as examples of use of pheromones is hardly representative as, in fact, the synthetic pheromones of these two species, particularly the latter, are not as attractive as in many other species and non-chemical factors are at least as important.  I think the authors just have to say that pheromones are now widely use in pest management for monitoring and control by mass trapping, lure-and-kill and mating disruption and give a reference such as Lucchi A, Benelli G  (2022) From Insect Pheromones to Mating Disruption: Theory and Practice. https://doi.org/10.3390/books978-3-0365-3178-6

Response: The authors thank the reviewers for their comments. We agree that the examples are quite specific, and have therefore removed the examples of the Emerald Ash Borer, and will include the reviewers addition with the reference.

The section on redbay ambrosia beetle seems unjustifiably long at nearly a page, perhaps driven by the interest of one of the contributors, and could be greatly shortened to allow inclusion of other examples.

Response: This section has now been condensed

Semiochemicals produced by yeasts are highlighted earlier and so it would be good to mention widespread use of microbial volatiles as lures for a range of insects, particularly fruit flies such as SWD.

In Section 6.3. it could be mentioned that one of the most significant and commercially-advanced uses of transgenic approaches is that of BioPhero using transgenic yeasts to produce pheromones on large scale commercially. Holkenbrink et al. (2020) https://doi.org/10.1016/j.ymben.2020.10.001

Response: The authors thank the reviewer for this suggestion. We have done some background reading on the topic, and agree that this is a great example to inculde.

Other relatively minor editorial comments are as follows.

L 171, 379, 621. (E)-β-caryophyllene

L209 and throughout. Standardise on E and Z rather than cis and trans in places

L451. “Behaviour of natural enemies” is less clunky

L479. the role of semiochemicals has been poorly explored [152], and similarly for PGPR (see [153]).

L514. As the infection develops, however, aphids are re-

L530. 6. Application of semiochemicals to improve sustainability in agriculture

L534. Predalure is a semiochemical lure, not pheromone lure

References. Insect names should be in italics?

Response: These amendments have been made.